# Characterizations of the Gut Bacteriome, Mycobiome, and Virome in Patients with Osteoarthritis

Changming Chen,[a] Yue Zhang,[b] Xueming Yao,[a] Shenghui Li,[b] Guangyang Wang,[c] Ying Huang,[a] Yutao Yang,[a] Aiqin Zhang,[b] Can Liu,[a] Dan Zhu,[a] Hufan Li,[a] Qiulong Yan,[c] Wukai Ma[a]

[a]Department of Rheumatology and Immunology, The Second Affiliated Hospital of Guizhou University of Traditional Chinese Medicine, Guiyang, China
[b]Puensum Genetech Institute, Wuhan, China
[c]Department of Microbiology, College of Basic Medical Sciences, Dalian Medical University, Dalian, China

Changming Chen and Yue Zhang contributed equally to this work. Author order was determined both alphabetically and in order of increasing seniority.

**ABSTRACT** The gut microbiota plays an essential role in the regulation of the immune system and the etiology of human autoimmune diseases. However, a holistic understanding of the gut bacteriome, mycobiome, and virome in patients with osteoarthritis (OA) remains lacking. Here, we explored the gut microbiotas of 44 OA patients and 46 healthy volunteers via deep whole-metagenome shotgun sequencing of their fecal samples. The gut bacteriome and mycobiome were analyzed using a reference-based strategy. Gut viruses were identified from the metagenomic assembled contigs, and the gut virome was profiled based on 6,567 nonredundant viral operational taxonomic units (vOTUs). We revealed that the gut microbiome (including bacteriome, mycobiome, and virome) of OA patients is fundamentally altered, characterized by a panel of 279 differentially abundant bacterial species, 10 fungal species, and 627 vOTUs. The representative OA-enriched bacteria included *Anaerostipes hadrus* (GENOME147149), *Prevotella* sp900313215 (GENOME08259), *Eubacterium_E hallii* (GENOME000299), and *Blautia* A (GENOME001004), while *Bacteroides plebeius* A (GENOME239725), *Roseburia inulinivorans* (GENOME 001770), *Dialister* sp900343095 (GENOME075103), *Phascolarctobacterium faecium* (GENOME233517), and several members of *Faecalibacterium* and *Prevotella* were depleted in OA patients. Fungi such as *Debaryomyces fabryi* (GenBank accession no. GCA_003708665), *Candida parapsilosis* (GCA_000182765), and *Apophysomyces trapeziformis* (GCA_000696975) were enriched in the OA gut microbiota, and *Malassezia restricta* (GCA_003290485), *Aspergillus fumigatus* (GCA_003069565), *and Mucor circinelloides* (GCA_010203745) were depleted. The OA-depleted viruses spanned *Siphoviridae* (95 vOTUs), *Myoviridae* (70 vOTUs), and *Microviridae* (5 vOTUs), while 30 *Siphoviridae* vOTUs were enriched in OA patients. Functional analysis of the gut bacteriome and virome also uncovered their functional signatures in relation to OA. Moreover, we demonstrated that the OA-associated gut bacterial and viral signatures are tightly interconnected, suggesting that they may impact disease together. Finally, we showed that the multikingdom signatures are effective in discriminating the OA patients from healthy controls, suggesting the potential of gut microbiota for the prediction of OA and related diseases. Our results delineated the fecal bacteriome, mycobiome, and virome landscapes of the OA microbiota and provided biomarkers that will aid in future mechanistic and clinical intervention studies.

**IMPORTANCE** The gut microbiome of OA patients was completely altered compared to that in healthy individuals, including 279 differentially abundant bacterial species, 10 fungal species and 627 viral operational taxonomic units (vOTUs). Functional analysis of the gut bacteriome and virome also revealed their functional signatures in relation to OA. We found that OA-associated gut bacterial and viral signatures were tightly interconnected, indicating that they may affect the disease together. The OA

Address correspondence to Qiulong Yan, qiulong1988@163.com, or Wukai Ma, walker55@163.com.

The authors declare no conflict of interest.

patients can be discriminated effectively from healthy controls using the multikingdom signatures, suggesting the potential of gut microbiota for the prediction of OA and related diseases.

**KEYWORDS** osteoarthritis, whole-metagenome shotgun sequencing, gut bacteriome, gut mycobiome, gut virome, microbiota dysbiosis

Osteoarthritis (OA) is a degenerative disease in middle-aged and older populations which is characterized by synovial inflammation, reactive hyperplasia of articular margins and subchondral bone, loss and degradation of articular cartilage, and even loss of mobility (1–3). Osteoarthritis is one of the most common chronic joint pain diseases in both developed and developing countries (1, 4), and it entails substantial social and medical burdens. Although the pathogenesis of OA remains largely unknown, exercise, diet, aging, obesity, strain, trauma, joint congenital abnormalities, joint deformity, and many other factors have been implicated as potential risk factors for this disease (1).

Interestingly, most of the above-mentioned potential risk factors of OA are related to the gut microbiota, which was considered the main pathogenic factor for OA (5), and therefore, a "gut-joint axis" concept was proposed (6). The human gut microbiota is composed of more than 4,000 bacterial species as well as a variety of fungi, viruses, phages, parasites, and archaea, which can synthesize various vitamins and amino acids, participate in the metabolism of sugar and protein, and promote the absorption of mineral elements (7–10). Gut microbiota dysbiosis can contribute to inflammation by inducing the production of proinflammatory cytokines by host immune cells and by the production of inflammatory bacterial metabolites (11). Several studies characterized the gut microbiota, explained the existence of low-grade intestinal inflammation in OA, and suggested a potential role for the microbiota in OA-related pain (12–14). The microbial metabolites could be also closely related to OA. For example, short-chain fatty acids (SCFAs) secreted by microorganisms might have a key role in maintaining bone homeostasis, downregulating the proinflammatory stimuli sustained by regulatory T-cells, and inhibiting bone resorption through direct inhibition of osteoclast activity (15). An increased intestinal permeability due to "leaky gut syndrome" led to the translocation of microbial products through the intestinal tight junctions, resulting in endotoxemia and consequently a proinflammatory status (16). Fecal microbiota transplantation from metabolically compromised donors accelerates osteoarthritis in mice (17), which established a direct connection between gut microbiota and OA. In addition, the potential role of viruses in joint disease, such as human endogenous retrovirus W (18), and Epstein-Barr virus (19), has also been initially and sporadically explored.

Although mycobiome and virome research of human diseases has developed rapidly in recent years, to the best of our knowledge, no study has examined the association among the gut bacteriome, mycobiome, and virome in OA patients. Elucidating this association might help clarify the role of gut microorganisms in the development of OA and contribute to potential translational opportunities for the prevention and treatment of OA.

## RESULTS

**Biodiversity and phylogenetic and functional compositions of the gut bacteriome.** Based on whole-metagenome shotgun sequencing, we obtained a total of 719.7 Gbp of high-quality nonhuman data (on average, 8.0 Gbp per sample) from the fecal samples of 44 osteoarthritis patients and 46 age- and body mass index (BMI)-matched healthy individuals (see Tables S1 and S2 in the supplemental material). To investigate the composition of gut bacteria and archaea (referred to as the bacteriome here), we mapped the metagenomic reads to the Unified Human Gastrointestinal Genome (UHGG) database (20) and obtained the bacteriome profiles of 5,187 taxa, including 17 phyla, 35 classes, 75 orders, 145 families, 392 genera, and 4,598 species. First, we performed the within-sample biodiversity analysis of the gut bacteriome at the species level. Rarefaction analysis revealed that the number of observed species was slightly

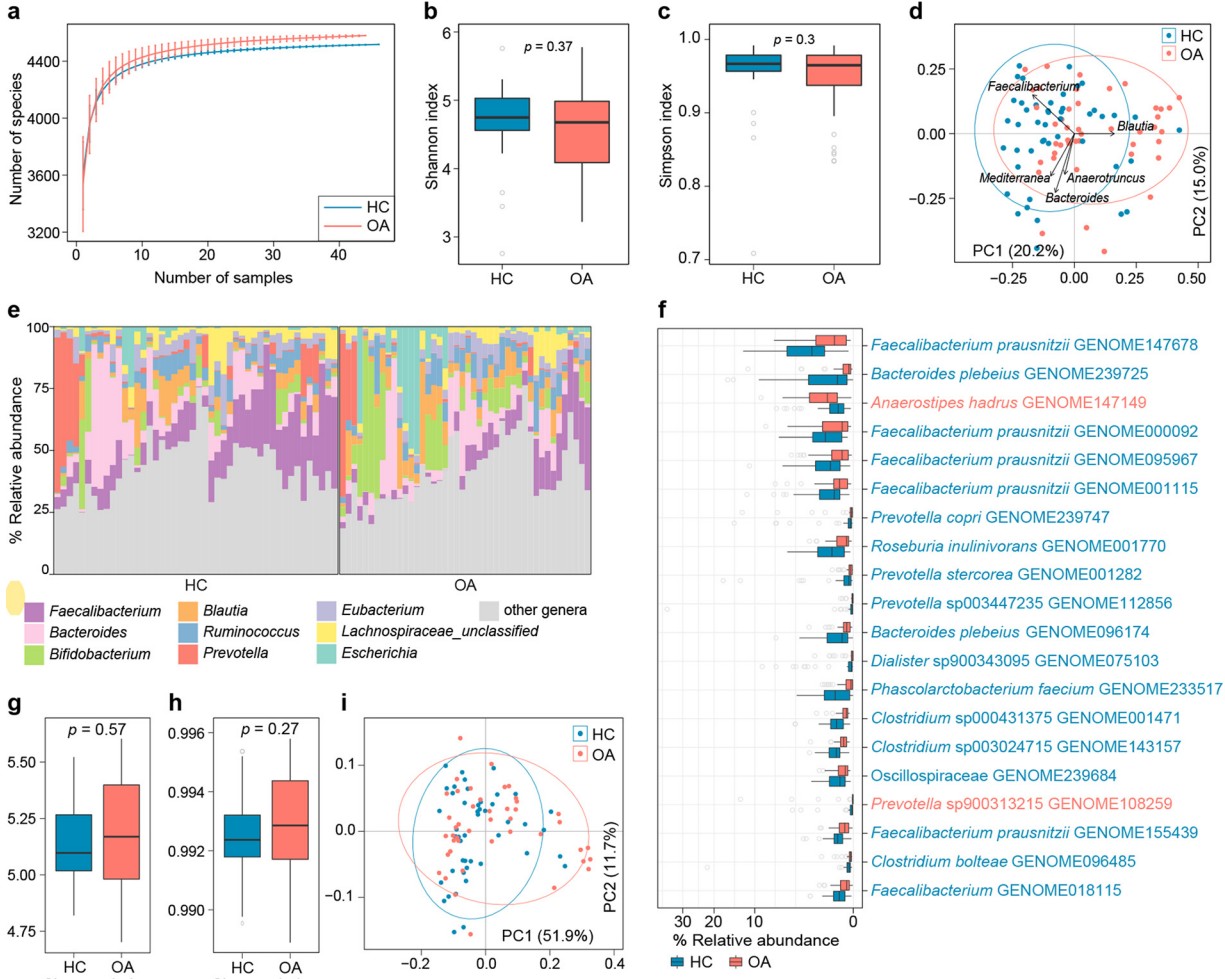

**FIG 1** Difference in gut bacteriome between OA patients and healthy controls. (a) Rarefaction curve analysis of the number of observed species for each group of samples. The number of species in different groups was calculated based on a randomly selected specific number of samples with 30 replacements, and the median and quartile values are plotted. (b and c) Box plots showing the Shannon diversity index (b) and the Simpson index (c) of gut bacteriomes that differ between the two groups. (d) PCoA of Bray-Curtis distance based on the composition of gut bacteriome, revealing the separation between the two groups. The location of samples (represented by nodes) in the first two principal coordinates is shown. Lines connect samples in the same group, and circles cover samples near the center of gravity for each group. The top five genera as the main contributors were plotted by their loadings in these two coordinates. (e) Composition of gut bacteriome at the genus level. (f) Box plot showing the representative differential gut bacterial species compared between the OA and HC groups. Species that are more abundant in patients and controls are colored red and blue, respectively. (g and h) Box plot showing the Simpson index (g) and Shannon diversity index (h) of gut functional composition that differ between the two groups. (i) PCoA of Bray-Curtis distance based on the gut functional composition, revealing the separations between two groups. For box plots, boxes represent the interquartile range between the first and third quartiles and median (internal line); whiskers denote the lowest and highest values within 1.5 times the range of the first and third quartiles, respectively; and nodes represent outliers beyond the whiskers. The significance level was calculated based on Student's *t* test.

but not significantly higher in the bacteriome of OA patients compared with that in healthy controls in the same sample size (Fig. 1a). Also, the Shannon diversity index and Simpson index were similar between OA and healthy control (HC) groups (Student's *t* test $P > 0.05$ for both indexes) (Fig. 1b and c). Next, we undertook a principal-coordinate analysis (PCoA) to further understand the differences in gut bacteriome between OA and HC individuals. Clear separations were shown between the OA and HC subjects (permutational multivariate analysis of variance [PERMANOVA] $P < 0.001$) (Fig. 1d), suggesting a substantial distinction in the overall gut microbial structure between them.

At the phylum level, the gut bacteriome of OA patients had markedly higher levels of *Actinobacteriota* (average abundance, 15.6% versus 6.4%; Wilcoxon rank-sum test $q < 0.001$) and *Proteobacteria* (7.8% versus 3.3%; $q = 0.02$) and lower levels of

*Firmicutes* (59.4% versus 68.1%; *q* = 0.03) than that of the HC subjects (Table S3). At the genus level, 8 genera, including *Anaerostipes*, *Bifidobacterium*, *Brachyspira*, and *Eggerthella*, were enriched in OA subjects compared to HC subjects, while 15 genera, such as *Faecalibacterium*, *Lachnoclostridium*, *Phascolarctobacterium*, and *Paraprevotella*, were enriched in the HC subjects (Table S4). In addition, we compared the gut bacteriome of OA and HC subjects at the species level. Two hundred seventy-nine species were identified with significant differences in relative abundance between the two cohorts (Wilcoxon rank-sum test *q* < 0.05), while 41 of these species were enriched in the OA subjects and 238 of them were enriched in the HC subjects (Fig. 1e; Table S5). The representative OA-enriched species included *Anaerostipes hadrus* (GENOME147149), *Prevotella* sp900313215 (GENOME108259), *Eubacterium_E hallii* A (GENOME000299), *Bifidobacterium* sp002742445 (GENOME027286), *Anaerostipes hadrus* A (GENOME000113), *Blautia* A (GENOME001004), *Catenibacterium* sp000437715 (GENOME001490), and *Bifidobacterium dentium* (GENOME096132), while the HC-enriched species included *Bacteroides plebeius* A (GENOME239725), *Roseburia inulinivorans* (GENOME001770), *Dialister* sp900343095 (GENOME075103), *Phascolarctobacterium faecium* (GENOME233517), and several members of *Faecalibacterium* (*Faecalibacterium prausnitzii* G [GENOME147678], *F. prausnitzii* K [GENOME095967], *F. prausnitzii* C [GENOME000092], and *F. prausnitzii* E [GENOME001115]) and *Prevotella* (*Prevotella stercorea* [GENOME001282], *Prevotella* sp003447235 [GENOME112856], and *Prevotella copri* A [GENOME239747]) (Fig. 1f).

Finally, we profiled the functions of the gut bacteriome of all fecal metagenomes using the HUMAnN3 algorithm (21), representing a total of 465 MetaCyc pathways for comparison analysis between the OA and HC subjects. Biodiversity analysis at the pathway level showed that the prokaryotic functional compositions of OA patients and HC subjects were similar in their Shannon and Simpson indexes (Fig. 1g and h). Consistent with the observation in the phylogenetic composition, the functional composition of the two groups at the PCoA plot also differed (PERMANOVA *P* = 0.04) (Fig. 1i). Fifteen of 465 pathways differed significantly between two cohorts (5 and 10 pathways were enriched in OA and HC subjects, respectively) (Table S6). The representative OA-enriched pathways included a partial tricarboxylic acid (TCA) cycle (PWY-5913), inosine-5′-phosphate biosynthesis III (PWY-7234), the C4 photosynthetic carbon assimilation cycle, NAD-dependent malic enzyme (NAD-ME) type (PWY-7115), and poly(glycerol phosphate) wall teichoic acid biosynthesis (TEICHOICACID-PWY), while the representative HC-enriched pathways included chorismate biosynthesis from 3-dehydroquinate (PWY-6163), purine ribonucleosides degradation (PWY0-1296), flavin biosynthesis (RIBOSYN2-PWY and PWY-6168), thiamine phosphate formation from pyrithiamine and oxythiamine (PWY-7357), pyrimidine deoxyribonucleosides salvage (PWY-7199), and biotin biosynthesis II (PWY-5005).

**Biodiversity and phylogenetic comparison of the gut mycobiome.** We profiled the gut fungal composition ("mycobiome") of all fecal metagenomes based on the available gut fungal genomes in the NCBI-RefSeq database (see Materials and Methods). The composition of 106 fungal species was generated and compared between the OA and HC groups. Analysis of fungal within-sample biodiversity revealed that the gut mycobiome was not significantly different between HC and OA groups with regard to the Shannon index and Simpson index (Student's *t* test *P* > 0.05) (Fig. 2a and b). However, consistent with the observation in the gut bacteriome, PCoA of the gut mycobiome also showed a remarkable distinction between the OA and HC groups (PERMANOVA *P* = 0.019) (Fig. 2c). At the genus level, the gut mycobiome of OA subjects was dominated by *Saccharomyces* (average abundance, 23.4%), *Cryptococcus* (average abundance, 15.9%), and *Candida* (average abundance, 11.6%), while that of the HC subjects was composed of *Saccharomyces* (average abundance, 29.4%), *Cryptococcus* (average abundance, 12.9%), and *Malassezia* (average abundance, 9.2%) (Fig. 2d). Of these, *Malassezia* and *Candida* significantly differed in abundance between the two cohorts (Wilcoxon rank-sum test *q* < 0.05) (Table S7). At the species level, 10 fungi were identified as differential species between the gut mycobiomes of OA and HC subjects (*q* < 0.05)

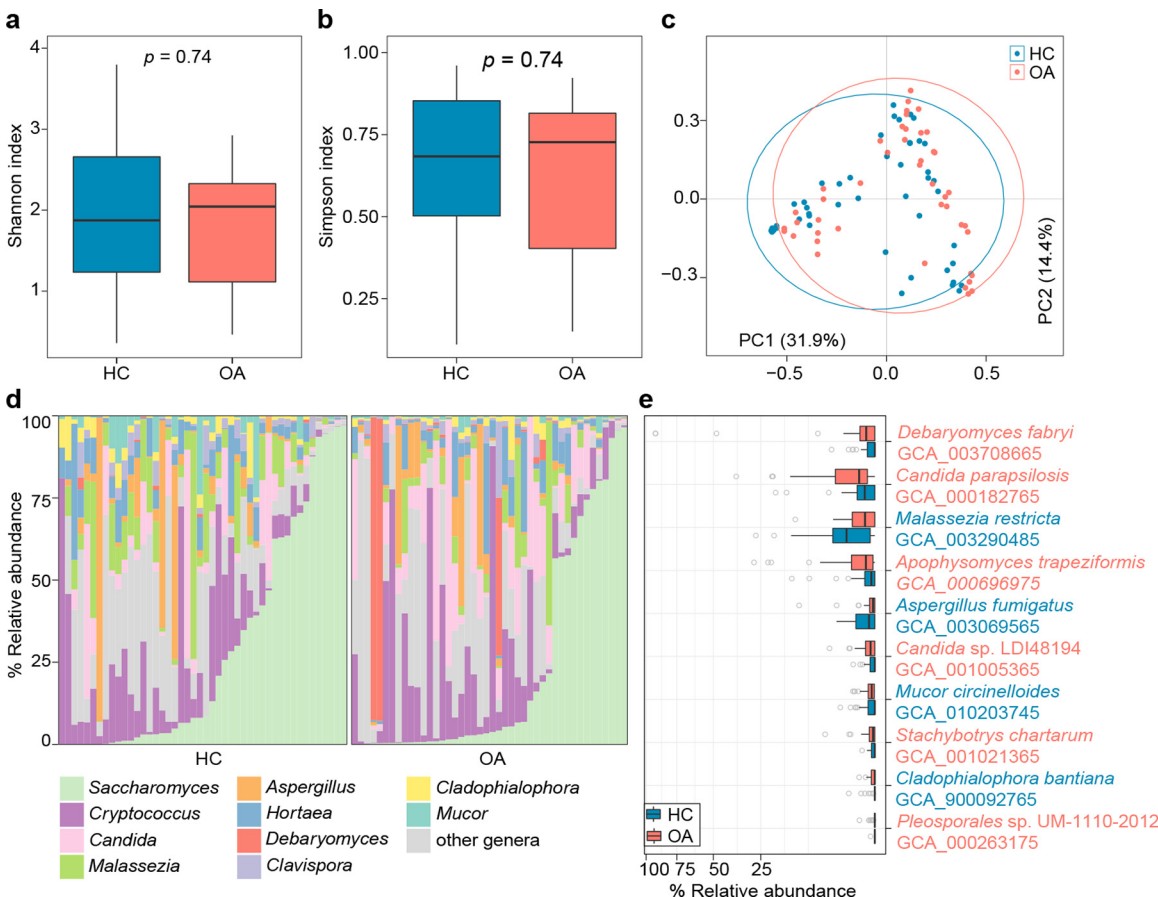

**FIG 2** Difference in gut mycobiome between OA patients and healthy controls. (a and b) Box plot showing the Shannon diversity index (b) and the Simpson index (c) of gut mycobiome that differ between two groups. The significance level was calculated based on Student's *t* test. (c) PCoA of Bray-Curtis distance based on the composition of the gut mycobiome, revealing the separation between the two groups. The location of samples (represented by nodes) in the first two principal coordinates is shown. Lines connect samples in the same group, and circles cover samples near the center of gravity for each group. (d) Composition of gut mycobiomes at the genus level. (e) Box plot showing the OA-associated gut fungal species compared between OA patients and healthy controls. Species that are more abundant in patients and controls are colored red and blue, respectively. For box plots, boxes represent the interquartile range between the first and third quartiles and median (internal line); whiskers denote the lowest and highest values within 1.5 times the range of the first and third quartiles, respectively; and nodes represent outliers beyond the whiskers.

(Fig. 2e). *Debaryomyces fabryi* (GenBank accession no. GCA_003708665), *Candida parapsilosis* (GCA_000182765), *Apophysomyces trapeziformis* (GCA_000696975), *Candida* spLDI48194 (GCA_001005365), *Stachybotrys chartarum* (GCA_001021365), and *Pleosporales* spUM-1110-2012 (GCA_000263175) were more abundant in the gut mycobiome of OA patients, while *Malassezia restricta* (GCA_003290485), *Aspergillus fumigatus* (GCA_003069565), *Mucor circinelloides* (GCA_010203745), and *Cladophialophora bantiana* (GCA_900092765) were more abundant in HC subjects.

**Construction of gut virus catalogue and comparison of gut virome.** To explore the gut viromes of OA and HC individuals, we obtained a total of 12,044 viral sequences (length, ≥5,000 bp) from the assembled contigs of all 90 fecal metagenomes (Table S2). These viral contigs were clustered into 6,567 viral operational taxonomic units (vOTUs) at 95% nucleotide similarity and 70% coverage. The length of this vOTU catalogue ranged from 5,000 bp to 546,697 bp, with an average length of 24,511 bp and an $N_{50}$ of 54,784 bp. Based on the estimation result by CheckV (22), 15.9% of these vOTUs were evaluated as complete or high-quality viral genomes, and 16.6% and 67.5% of them were medium- and low-quality genomes, respectively (Fig. 3a). Noticeably, only 53.6% (*n* = 3,523) of 6,567 vOTUs were shared with the currently known collections of human gut virome, including the Gut Virome Database (23), Gut

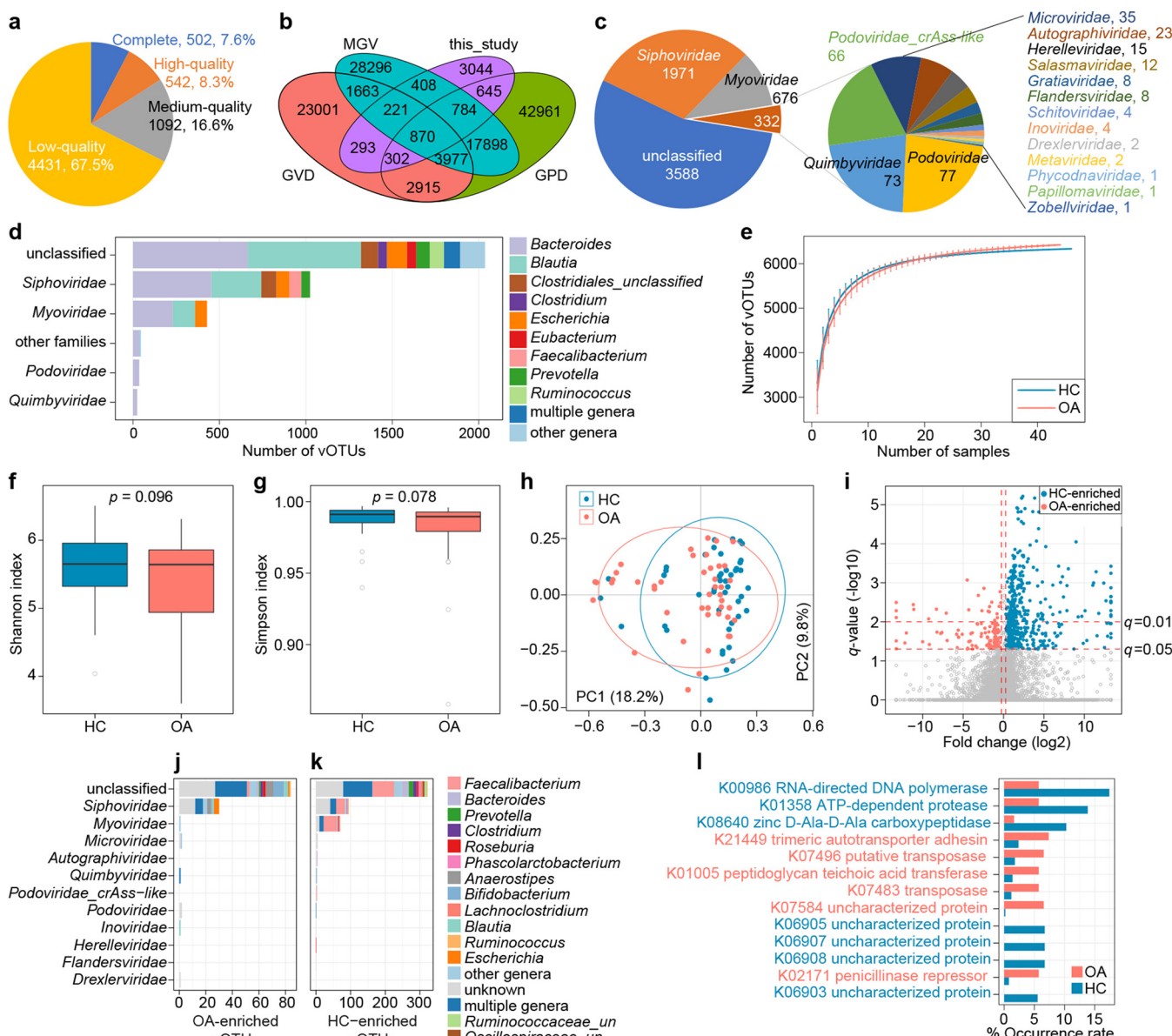

**FIG 3** Characteristics of the gut virus catalogue and gut virome. (a) Pie plot showing the proportions of complete, high-quality, medium-quality, and low-quality vOTUs in the nonredundant virus catalogue. (b) Venn plot showing the overlap of the current virus catalogue and the other three gut virus catalogues. (c) Pie plot showing the family-level taxonomic annotation of the nonredundant virus catalogue. (d) Bacterial host assignment of the nonredundant virus catalogue. (e) Rarefaction curve analysis of the number of observed vOTUs on each group of samples. The number of species in different groups was calculated based on a randomly selected specific number of samples with 30 replacements, and the median and quartile values were plotted. (f and g) Box plots showing the Shannon diversity index (f) and the Simpson index (g) of gut viromes that differ between the two groups. Boxes represent the interquartile range between the first and third quartiles and median (internal line); whiskers denote the lowest and highest values within 1.5 times the range of the first and third quartiles, respectively; and nodes represent outliers beyond the whiskers. The significance level was calculated based on Student's t test. (h) PCoA of Bray-Curtis distance based on the composition of the gut virome, revealing the separation between two groups. The location of samples (represented by nodes) in the first two principal coordinates is shown. Lines connect samples in the same group, and circles cover samples near the center of gravity for each group. (i), Volcano plot shows the fold change versus q values for all vOTUs. The x axis shows the ratio of vOTU abundance in OA patients compared with that in healthy controls. The y axis shows the q value ($-\log_{10}$ transformed) of a vOTU. The vOTUs that were enriched in OA and HC subjects are shown with red and blue points, respectively. (j and k), Bar plots show the taxonomic distribution of OA-enriched vOTUs (j) and HC-enriched vOTUs (k). (l) Occurrence rate of 10 differential KOs that differed in frequency between the OA-enriched and HC-enriched vOTUs. KOs that are more abundant in patients and controls are colored red and blue, respectively.

Phage Database (24), and Metagenomic Gut Virus catalogue (25) (Fig. 3b), highlighting the considerable novelty of our vOTU catalogue.

Of 6,567 vOTUs, 45.4% could be robustly assigned to a known viral family. *Siphoviridae* (30.0%; n = 1,971) and *Myoviridae* (10.3%; n = 676) dominated the vOTUs that were classified, and other representatives included *Podoviridae*, *Quimbyviridae*, crAss-like *Podoviridae*,

*Microviridae, Autographiviridae, Herelleviridae, Salasmaviridae,* and so on (Fig. 3c). Moreover, 54.6% of the vOTUs could be assigned into one or more bacterial hosts based on their homology to genome sequences or CRISPR spacers of the aforementioned 4,644 gut prokaryotic species from the UHGG database. The most common identifiable hosts were members of *Firmicutes* (mainly *Blautia, Clostridium, Eubacterium,* and *Faecalibacterium*), *Bacteroidota* (mainly *Bacteroides* and *Prevotella*), and *Escherichia* (Fig. 3d).

Rarefaction analysis showed that the tendencies of accumulative curves of the OA and HC groups were similar (Fig. 3e). Comparisons of the Shannon diversity index and Simpson index also revealed that the within-sample diversities of the two groups were similar (Student's *t* test $P > 0.05$) (Fig. 3f and g).

PCoA showed that the gut viromes of OA patients and healthy controls were significantly different at the vOTU level (Fig. 3h), with a PERMANOVA *P* of <0.001. We then compared the composition of gut virome between the two groups at the vOTU level. In total, 627 vOTUs were identified with significant differences in relative abundances between OA patients and healthy controls ($q < 0.01$); 122 vOTUs of these were enriched in OA patients, and 505 were depleted (Fig. 3i; Table S8). The OA-enriched vOTUs spanned 7 known viral families, which contained 30 *Siphoviridae* viruses, whereas the HC-enriched vOTUs spanned 10 viral families, containing 95 *Siphoviridae*, 70 *Myoviridae*, and 5 *Microviridae* viruses (Fig. 3j and k).

Finally, we predicted 39,677 genes from the 627 differential vOTUs and annotated 27.3% of them based on the KEGG (Kyoto Encyclopedia of Genes and Genomes) database. These annotated genes were assigned to 2,306 KEGG orthologs (KOs) for further analysis. Thirteen KOs occurred at significantly different frequencies between the OA-enriched and HC-enriched vOTUs (Wilcoxon rank-sum test $q < 0.05$) (Fig. 3l; Table S9). Six of these KOs were more frequent in OA-enriched vOTUs, such as genes encoding the trimeric autotransporter adhesin K21449, transposases K07496 and K07483, penicillinase repressor K02171, and peptidoglycan teichoic acid transferase K01005, whereas 7 KOs, including genes for the RNA-directed DNA polymerase K00986, ATP-dependent protease K01358, and zinc D-Ala-D-Ala carboxypeptidase K08640, were enriched in the HC-enriched vOTUs.

**Correlations among gut bacteriome, mycobiome, and virome.** To explore the relationship among OA-associated gut bacterial, fungal, and viral signatures, we performed a correlation analysis between 279 differential bacterial species, 10 differential fungal species, and 627 differential vOTUs. Using an absolute Spearman correlation coefficient threshold of 0.6, we generated a large coabundance network with 11,082 coabundance relationships between 200 bacterial species and 418 vOTUs (Fig. 4a). However, no strong coabundance correlation was found between gut fungi and bacteria or viruses. vOTUs belonging to *Siphoviridae* were primarily connected to members of *Bacteroidota* and *Firmicutes*, while vOTUs belonging to *Myoviridae* were connected to a wide range of bacteria belonging to *Actinobacteriota, Firmicutes,* and *Bacteroidota*. Several bacterial taxa, including *Oscillospiraceae, Ruminococcaceae, Bacteroidaceae,* and *Faecalibacterium,* were frequently correlated with a large number of viruses, while some vOTUs were connected to the highest number of bacteria (Fig. 4b and c); these findings suggest potential central roles of these bacteria and viruses in the network. In addition, unlike the *Myoviridae* members, some viruses seem to act on disease independently of bacteria, such as the vOTUs belonging to *Microviridae* (Fig. 4d and e); their roles in disease also need further study.

**Classification of osteoarthritis based on multikingdom signatures.** Finally, we evaluated the performance of gut multikingdom signatures to identify OA status using the random forest model. The models that were trained based on the bacterial, fungal, and viral signatures obtained discriminatory powers of the area under the receiver operator characteristic (ROC) curve (AUC) of 0.947, 0.729, and 0.953, respectively (Fig. 5a). Likewise, a model trained based on the relative abundances of all differential bacteria, fungi, and vOTUs achieved an AUC of 0.955 (95% confidence interval, 0.919 to 0.991). Several bacteria, including *Lachnospiraceae* UBA9502 (GENOME085442), *Prevotella* sp002265625 (GENOME271908), *Prevotella* sp900313215 (GENOME108259), and *Lachnospiraceae* CAG-81 (GENOME139200), and some vOTUs featured the highest score for the discrimination of OA patients and healthy controls (Fig. 5b).

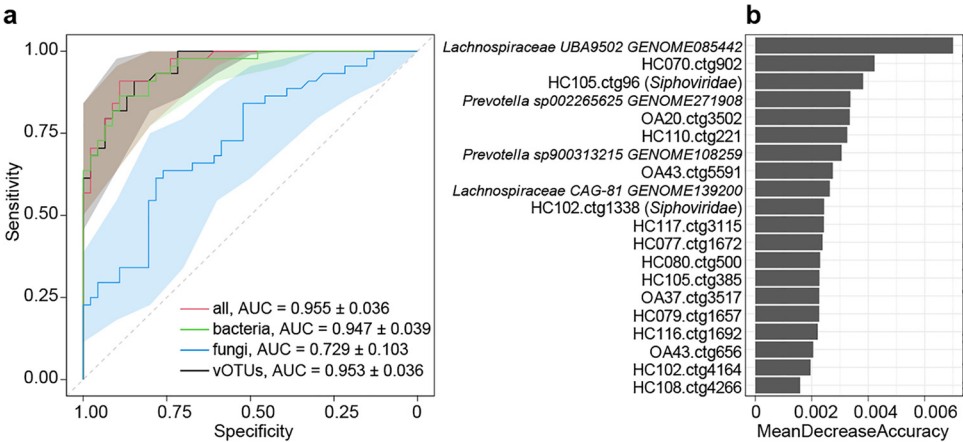

**FIG 4** Correlation analysis among gut bacteriome, mycobiome, and virome. (a) Network showing the correlations between gut bacteriome and virome. Bacterial species and vOTUs are grouped at the family level. (b and c) Bar plots showing the top 20 gut bacteria (b) and vOTUs (c) with the largest number of connections in the gut bacteriome-virome network. The taxonomic information on vOTUs is given inside the bars. (d and e) Pie plots showing the taxonomic distribution of bacterium-dependent (d) and bacterium-independent (e) vOTUs.

## DISCUSSION

As a chronic joint disease, osteoarthritis involves inflammation level and elevated levels of lipopolysaccharide (LPS) and is frequently associated with obesity and metabolic syndrome, which may significantly impact the quality of life of patients (14). The

**FIG 5** Classification of osteoarthritis status by the abundances of gut multikingdom signatures. (a) ROC analysis for classification of osteoarthritis status using the gut bacterial, fungal, and viral signatures. (b) The 20 most discriminant signatures in the model classifying OA patients and healthy controls. The bar lengths indicate the importance of the variable.

gut microbiota, which mainly consists of the bacteriome, mycobiome, and virome, makes an important contribution to human metabolic homeostasis (26) and is closely related to OA. However, systematic characterizations of the gut bacteriome, mycobiome, and virome of OA patients have been missing. Herein, we analyzed 90 fecal samples (OA, $n$ = 44; HC, $n$ = 46) using deep whole-metagenome shotgun sequencing and systematically compared the multikingdom microbial compositions between patients and controls. Our results showed that, though the species diversity had no statistical difference, there were significant differences in the abundance of 279 bacterial species, 10 fungal species, and 627 viruses between two groups. Identification of these gut microbial signatures may facilitate further mechanistic studies of OA and related diseases.

Specifically, the gut bacteriome of OA patients was significantly enriched in bacterial species, including *Anaerostipes hadrus* (GENOME147149), *Prevotella* sp900313215 GENOME108259), *Eubacterium_E hallii* E (GENOME000299), *Blautia* A (GENOME001004), and, in contrast, decreased in *Bacteroides plebeius* A GUT (GENOME239725), *Roseburia inulinivorans* (GENOME001770), and *Prevotella copri* (GENOME239747) and several subspecies of *Faecalibacterium prausnitzii* compared with that of HC subjects. High-abundance *Eubacterium* was also found in knee osteoarthritis in a rat model (27), suggesting potential involvement of this bacterium.

Notably, most of the decreased gut bacteria in OA patients are next-generation probiotics with the capability of managing metabolic diseases (28, 29) and producing short-chain fatty acids (SCFAs) (30, 31). For example, *Faecalibacterium prausnitzii* is a typical butyrate-producing bacterium in the gut (31), and butyrate could inhibit the inflammatory response through the NF-$\kappa$B signaling pathway (32) and promote human health. *Faecalibacterium prausnitzii* and *Roseburia inulinivorans* were significantly reduced in the gut microbiotas of Crohn's disease and type 1 diabetes patients compared to those in healthy individuals (33, 34). Consistently, data from the Xiangya Osteoarthritis Study showed that a lower relative abundance of the genus *Roseburia* was associated with symptomatic hand OA (35). In addition, Wang et al. found that *Prevotella* was significantly decreased in OA individuals and that it serves as a microbial biomarker of OA (36), suggesting a critical but underexplored role of this taxon in OA etiology. In addition, *Bacteroides plebeius* enabled other *Bacteroides* species to access the sulfated arabinogalactan proteins, providing a route for introducing privileged nutrient utilization into probiotic and commensal organisms that could improve human health (37).

In the gut mycobiome, *Saccharomyces*, *Candida*, *Cryptococcus*, and several other genera (e.g., *Malassezia* and *Mucor*) were the most dominant genera in all investigated samples. Fungi such as *Cryptococcus* and *Mucor* are not residents of the gut microbiota but may be linked to pathogenic stages in mostly immunocompromised humans (38, 39). Similarly, we compared the gut fungal profiles between OA patients and healthy controls and found that *Debaryomyces fabryi* (GenBank accession no. GCA_003708665), *Candida parapsilosis* (GCA_000182765), *Apophysomyces trapeziformis* (GCA_000696975), and *Candida* spLDI48194 (GCA_001005365) were enriched in the gut mycobiome of OA subjects compared with that of healthy subjects, while *Malassezia restricta* (GCA_003290485) and *Aspergillus fumigatus* (GCA_003069565) were depleted.

*Candida* fungi are opportunistic pathogens usually existing on the surface of the human body and in the oral cavity. When the normal microbiota is dysregulated or the resistance is reduced, some *Candida* species (e.g., *C. albicans*) can invade the body and cause infections in mucosa, viscera, and the central nervous system (40). Reports of *Candida* arthritis are rare, mostly in high-risk groups after arthroplasty, with diabetes and immunosuppressant administration. The occurrence of *Candida* arthritis in immunologically normal, nonpostoperative patients may be associated with multiple injections in the joint cavity and frequent intraarticular glucocorticoid applications (41). *Apophysomyces*, an unusual cause of mucormycosis (42), is known to cause cutaneous fungal infections such as septic arthritis (a rare clinical manifestation), particularly after penetrating trauma (43). *Apophysomyces trapeziformis* was also detected in cases of

necrotizing soft tissue infection (44, 45). On the other hand, *Malassezia* is the main culprit in skin diseases such as dandruff and seborrheic dermatitis. In addition, *Malassezia* exists as a commensal and benignly interacts with keratinocytes and the immune system in healthy skin (46).

With regard to the virome, 122 and 505 vOTUs were enriched in the gut viral community of patient and control cohorts, respectively. The majority of these viruses belong to the *Caudovirales*, including members of the *Siphoviridae*, *Myoviridae*, and *Podoviridae* (47). In addition, a small proportion of OA-depleted vOTUs were members of *Microviridae*, which are isoaxial phages lacking a tail structure and are limited to Gram-negative bacteria such as enterobacteria (48). Currently, our understanding of gut phages and their effects on human health and disease is in the early stages. Although no pathogenic association has been established between gut phages and disease, given the ability of phages to regulate host bacteria, further research is needed to determine whether gut phage-mediated microbiota changes lead to development of or result from the disease (49). Studies of gut phages in inflammatory bowel disease (IBD) patients showed that the relative abundance of *Caudovirales* is higher than that of *Microviridae* in IBD patients, and the composition of *Caudovirales* families is different from that in healthy individuals (50). Moreover, there was a significant association between IBD and rheumatoid arthritis (RA), and early-RA patients were susceptible to chronic inflammation (51). Among the genes carried by the OA-associated viruses, we found a higher frequency of the OA-enriched vOTUs containing genes encoding the peptidoglycan teichoic acid transferase (K01005), in agreement with the observations in the gut virome of RA patients (52). Virus-encoded peptidoglycan teichoic acid transferase links to the degradation of the peptidoglycans of the bacterial cell wall during infection (53), which may lead to increased levels of proinflammatory bacterial debris, thus affecting local innate immune responses and the mucosal immune system (54, 55). These results suggested that the different gut microbiological compositions of OA and HC viromes may influence host health.

Based on bacterial 16S rRNA gene sequencing, Wang et al. identified 7 optimal microbial biomarkers at the genus level as a panel, including *Gemmiger*, *Klebsiella*, *Akkermansia*, *Bacteroides*, *Prevotella*, *Alistipes*, and *Parabacteroides*, to build the random forest model and achieved an AUC of 0.834 for classifying overweight people at risk for OA in healthy individuals (36). This is an encouraging development; however, because it is not limited to bacteria, we obtained a more accurate and higher AUC of 0.955 for OA discrimination using the combined gut bacterial, fungal, and viral signatures.

In summary, we have for the first time systematically characterized the gut bacteriome, mycobiome, and virome in OA patients via metagenomic sequencing of their fecal samples. Compared with the gut microbiome of healthy individuals, that of OA patients was completely altered, being characterized by a panel of 279 differentially abundant bacterial species, 10 fungal species, and 627 vOTUs. In addition, a functional analysis and a determination of the signatures of the OA-associated gut microbiota and multikingdom signatures in OA patients were performed, suggesting the potential effects of the gut microbiota in OA. Our research will provide insights that may be useful for future mechanistic and clinical intervention studies.

## MATERIALS AND METHODS

**Ethics statement, subjects, and sample collection.** The research protocol was approved by the Medical Ethical Committees of the Second Affiliated Hospital of Guizhou University of Traditional Chinese Medicine. All subjects who participated in this research provided written informed consent.

Forty-four osteoarthritis patients admitted to the Department of Rheumatology and Immunology, Second Affiliated Hospital of Guizhou University of Chinese Medicine, China, between August 2020 and August 2021 were enrolled in this study. All the patients fulfilled the guidelines for the diagnosis and treatment of osteoarthritis in China (2019 edition) (56). The disease activity was assessed according to the Western Ontario and McMaster Universities Arthritis Index (WOMAC) (57, 58) using fecal samples obtained before the patients were administered glucocorticoids and immunosuppressive agents. The following groups were excluded: (i) volunteers with other autoimmune diseases and patients who received any anti-inflammatory treatment within 1 month before sampling; (ii) OA patients and healthy subjects who received antibiotics, antifungals, or probiotic treatment within 1 month; and (iii)

individuals with excessive drinking habits and those who had drunk sour milk within 1 week. Forty-eight age-matched and body mass index (BMI)-matched healthy subjects (Table S1) without arthralgia, heart failure, renal failure, or autoimmune disease and free from other inflammatory conditions were recruited based on records available from the Department of Medical Examination Center, Second Affiliated Hospital of Guizhou University of Chinese Medicine.

Fecal samples from all individuals were self-collected after defecation at the hospital and immediately placed on dry ice. Then the samples were transferred to the laboratory, divided into two parts, and put into two frozen tubes. All the fecal samples were transferred to −80°C for storage.

**DNA extraction and whole-metagenome shotgun sequencing.** Fecal samples were collected with sterile feces collection containers and stored rapidly at −80°C until use. The total DNA of fecal samples (170 mg per sample) was extracted using the Tiangen fecal DNA extraction kit (Tiangen, China) according to the manufacturer's instructions. DNA concentration and purity were determined by NanoDrop2000 and Qubit 4.0. Total DNA was fragmented using Covaris M220 (Gene Company Limited, China). For each DNA sample, we constructed a 150-bp paired-end library with an insert size of approximately 350 bp. All libraries were barcoded and pooled to perform whole-metagenome shotgun sequencing on the Illumina NovaSeq platform. Initial base calling of the metagenomic data set was performed based on the system default parameters under the sequencing platform. The raw sequencing reads for each sample were independently processed for quality control using fastp (59). fastp processed the raw sequencing reads by trimming the low-quality ($Q < 30$) bases at the end of reads and filtering N-containing, adapter-contaminated, or short (<90-bp) reads to generate the high-quality reads. The human reads were removed from the high-quality reads based on their Bowtie2 (60) alignment to the human reference genome (GRCh38).

**Analyses of gut bacteriomes.** The gut bacteriome composition of fecal samples was profiled based on the extensive UHGG database (9) and was found to comprise 204,938 nonredundant genomes from 4,644 gut prokaryotes. The metagenomic reads for samples of OA patients and HC were aligned against the UHGG database to generate the gut bacteriome profiles. Reads that mapped to the bacterial rRNA/tRNA gene sequences were dismissed. Relative abundances of 4,644 prokaryotic species were calculated by normalizing for each sample, and the relative abundances at the phylum and genus levels were obtained by summing the abundances of species from the same taxa. The functional composition of the fecal metagenomes was obtained by using the HUMAnN3 algorithm (21), a method that efficiently and accurately profiled the abundance of microbial metabolic pathways and molecular functions directly from the metagenomic sequencing data set. The Shannon and Simpson diversity indexes for the taxonomic composition were calculated based on the relative abundance profile at the species level using the vegan package in the R platform, while those of the functional composition were calculated at the MetaCyc pathway level.

**Analyses of gut mycobiomes.** To profile the gut mycobiome composition for the fecal metagenomic samples, we downloaded the available fungal genomes from the National Center for Biotechnology Information (NCBI) RefSeq database.

The metadata for genomes were obtained from the NCBI BioSample database, and only fungal strains that were isolated or sourced from human feces and/or digestive tract specimens were included. The genomes of 1,503 gut fungi, corresponding to 106 species, were used as the fungal reference. Then, the high-quality nonhuman metagenome reads for each sample were aligned with the fungal genome references to generate the gut fungal profiles. Reads that mapped to the fungal rRNA/tRNA gene sequences were dismissed. To avoid potential contamination from other gut microbes (i.e., bacteria, archaea, and viruses), the reads that mapped to fungal genomes were then aligned against (i) all bacterial, archaeal, or viral sequences extracted from the NCBI NT database and (ii) 4,644 prokaryotic genomes from the UHGG database (9), and the contaminating reads thus identified were removed. Relative abundances of 106 fungal species were calculated by normalizing for each sample, and the relative abundances at the family and genera levels were obtained by summing the abundances of species from the same taxa. The Shannon and Simpson diversity indexes for the gut mycobiome composition were calculated based on the abundance profile at the species level.

**Identification of viral sequences and analyses of the gut virome.** High-quality metagenomic reads were used for *de novo* assembly via MEGAHIT (61) with a broad range of k-mer sizes (–k-list 21,41,61,81,101,121,141). In this study, we identified the viral sequences from the metagenomic assembled contigs following the methodology developed in several recent works (10, 23–25). For each sample, contigs of >5,000 bp were used for virus prediction by three approaches: CheckV (criterion: number of viral genes > number of microbial genes) (22), VIBRANT (62), and DeepVirFinder (criteria: score > 0.9 and $P$ value < 0.01) (63). Contigs that were predicted as viral sequences by any of these approaches were collected and further estimated for quality using CheckV. Only contigs of >10,000 bp or of medium or high quality were included for further analysis. The identified viral contigs were dereplicated at 95% sequence similarity and over 70% coverage to generate the vOTUs.

Putative proteins of viral sequences were predicted using the Prodigal algorithm (64) with the parameter -meta. The vOTUs were taxonomically annotated based on the method combining the GenomeNet Virus-Host Database (Virus-Host DB) (65) and the vConTACT2 pipeline (66). Briefly, protein-coding genes of vOTUs were aligned against a protein database combined from the Virus-Host DB, vConTACT2 references, and the viral proteins of crAss-like phages (67) and several candidate gut viruses from the study by Benler et al. (68). Protein alignment was performed using DIAMOND with the options –query-cover 50 –identity 30 –top 40 –score 50. A virus was considered part of a known viral family if 25% of its genes were assigned to that family. To search for the potential bacterial hosts of viruses, the CRISPR spacers in the bacterial genomic sequence of UHGG genomes were predicted using MinCED

(parameter -minNR 2) (69), and then the spacers were compared in a BLAST search to the viral sequences (blastn-short mode and bit score of >50) to identify the phage-bacterium host pairs. The matching bacterial host of the viral sequence was summarized at the genus level. To avoid ambiguity, the bacterial genus producing the highest number of spacers hits was considered the primary host. Functional annotation of vOTUs was performed based on the KEGG (70) database using DIAMOND (71).

**Correlation network analysis.** We performed a correlation analysis among the bacterial species, fungal species, and vOTUs using Spearman's rank correlation coefficient. For each pair of microbes, a correlation coefficient was calculated based on the relative abundances after adjusting for individuals' gender, age, and BMI. Only intermicrobe correlation coefficients greater than 0.6 (positive) or less than −0.6 (negative) were regarded as indicating strong correlations and included for analysis. The correlation network was visualized using Cytoscape v3.8.2 (72).

**Statistical analyses.** Statistical analyses were implemented on the R v4.0.1 platform. PCoA of Bray-Curtis distances was performed using the vegan package. PERMANOVA was carried out with the adonis function of the vegan package, and the adonis $P$ value was generated based on 1,000 permutations. Student's $t$ test and the Wilcoxon rank-sum test were used to measure statistical differences in the diversity and taxonomic levels, respectively, between the two cohorts. The $q$ values were used for multiple-testing correction and generated by the Benjamini-Hochberg procedure. A $P$ value (for a single test) or $q$ value (for multiple testing) of less than 0.05 was considered statistically significant. Random forest models were trained using the randomForest package (1,000 trees) to distinguish between OA patients and healthy controls by using the abundance profiles of the differential bacteria, fungi, and vOTUs.

**Data availability.** The raw metagenomic sequencing data set for this study has been deposited in the European Nucleotide Archive (ENA) at EMBL-EBI under accession number PRJEB52499 (https://www.ebi.ac.uk/ena/data/view/PRJEB52499). All other data supporting the findings of the study are available in the paper and supplemental material or from the corresponding authors upon request.

## SUPPLEMENTAL MATERIAL

Supplemental material is available online only.
**SUPPLEMENTAL FILE 1**, XLSX file, 0.01 MB.
**SUPPLEMENTAL FILE 2**, XLSX file, 0.02 MB.
**SUPPLEMENTAL FILE 3**, XLSX file, 0.01 MB.
**SUPPLEMENTAL FILE 4**, XLSX file, 0.01 MB.
**SUPPLEMENTAL FILE 5**, XLSX file, 0.04 MB.
**SUPPLEMENTAL FILE 6**, XLSX file, 0.01 MB.
**SUPPLEMENTAL FILE 7**, XLSX file, 0.01 MB.
**SUPPLEMENTAL FILE 8**, XLSX file, 0.1 MB.
**SUPPLEMENTAL FILE 9**, XLSX file, 0.1 MB.

## ACKNOWLEDGMENTS

C.C., Y.Z., Q.Y., and W.M. contributed to conception and design of the study. C.C. and Y.Z. wrote the manuscript. Q.Y. and W.M. led the writing (review and editing). Y.H., Y.Y., C.L., D.Z., and H.L. collected the samples and information. X.Y., S.L., G.W., and A.Z. performed the data analysis. All authors contributed to the article and approved the submitted version.

We declare that there is no conflict of interests.

This work was supported by grants from the Science and Technology Program of Guizhou Province (Guizhou Scientific Foundation–Platform and talent [2020]2202), the Science and Technology Program of Guizhou Province (Guizhou Scientific Foundation–Support [2020]4Y155), the National Natural Science Foundation of China (81760907 and 81902037), and the Scientific Research Project of the Second Affiliated Hospital of Guizhou University of Traditional Chinese Medicine (GZEYK-B[2021]2).

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
