## [Reviewer comments · Microbiology Spectrum]

Microbiology Spectrum

Characterizations of the gut bacteriome, mycobiome, and virome in patients with osteoarthritis

Changming Chen, Yue Zhang, Xueming Yao, Shenghui Li, Guangyang Wang, Ying Huang, Yutao Yang, Aiqin Zhang, Can Liu, Dan Zhu, Hufan Li, Qiulong Yan, and Wukai Ma

Corresponding Author(s): Wukai Ma, The Second Affiliated Hospital of Guizhou University of Traditional Chinese Medicine

Review Timeline:

Submission Date:	May 8, 2022
Editorial Decision:	July 9, 2022
Revision Received:	September 8, 2022
Accepted:	September 13, 2022

Editor: Florence Doucet-Populaire

Reviewer(s): The reviewers have opted to remain anonymous.

Transaction Report:

DOI: <https://doi.org/10.1128/spectrum.01711-22>

July 9, 2022

Prof. Wukai Ma
The Second Affiliated Hospital of Guizhou University of Traditional Chinese Medicine
Department of Rheumatology and Immunology
Feishan Street No.83
Guiyang, Guizhou
China

Re: Spectrum01711-22 (Characterizations of the gut bacteriome, mycobiome, and virome in patients with osteoarthritis)

Dear Prof. Wukai Ma:

Link Not Available

Sincerely,

Florence Doucet-Populaire

Journals Department
Reviewer comments:

Reviewer #1 (Comments for the Author):

1. Improper use of italics or non-italic in the strain name.
2. In this study, criteria for judging significance level should be unified.
3. The lack of the icon for OA and HC in Figure 1f and Figure 2e.
4. Figure 2d should be reconfirmed to describe relative abundance at the genus or family level.
5. The criterion for clustering viral contigs into vOTUs should be the same in the context.
6. Figure 3i and Figure 3l overlapped.
7. The color of the same family should be the same in Figure 3c, 3j and 3k.

8. The lack of the figure note for Figure 3I.
9. How was sample size calculated?

Reviewer #2 (Comments for the Author):

The manuscript is well written and shows extensive analysis of the microbiome found in patients with Osteoarthritis (OA) fecal samples. However, the sampling and analysis methods made are not correct for describing microbial patterns for AO. Then the results and conclusions are not valid.

The objective is described as elucidating associations of the gut microbiome and virome with OA. However, the authors also mentioned biomarkers as conclusions. Therefore, they should clarify the intentions of this work since the establishment of biomarkers requires different confirmations.

The authors show some flaws in technical criteria, including:

- They should avoid using "gut microbiome" since the whole sampling and analysis was based on fecal samples. They should use "fecal microbiome" instead. One can speculate that the fecal microbiome might also be present in the gut, but not that the whole gut microbiome is represented on the fecal sample.
-
- The authors should use "microbiome" to refer to bacteriome and mycobiome. However, using "microbiome" in addition to mycobiome is confusing and incorrect.
- Gut bacteriome analysis: It should be more accurate to use a complete prokaryotes database, for example, Silva or Greengenes. The authors are limiting the results to the known gut bacteriomes.
- Mycobiome: again, the fungal assignment is limited to a selection of 106 genomes. This made the study completely biased.
- It is not explained how the metagenomes were assembled.
- The cohort seems too short for this analysis and to support the conclusions. The variability in relative diversity profiles is high in all the analyses. Establishing a dysbiosis or microbiome profile from 20 patients is not possible. Then, a larger group of patients will be necessary to confirm the profile patterns. Or at least statistically prove that this sample is enough for some of the dysbiosis events.
- Also, the authors made a Spearman correlation of the microbiome with the OA condition, but they did not clarify other potential correlations of these results. For example, might it be an influence of ethnicity? Other risk factors? Diet? Age? How can they conclude an association without knowing other potential drivers of the differential microbiome between the cohorts? The baseline cohort characteristics need to be established and discard significant differences among them to present valid results.

Reviewer #3 (Comments for the Author):

Chen et al, pursued a very interesting study of comparing gut microbiome of health subset and individuals having osteoarthritis using whole genome sequencing. They found no significant difference in terms of individual microbial population. However, they report significant difference in abundance of identified genera suggesting it would help for developing future biomarkers to identify people with osteoarthritis. This is a very novel idea even though it does not address how to achieve such prediction. They only looked at people already having osteoarthritis and purpose of prediction for future risk is not possible. In addition, their sample size is too small for most of the statistics to accurately predict and the claims based on these should be toned down as significant observations. Authors are not highlighting the patient population with age group. What they observed is a change in the abundance, not the species diversity/variation. The authors conclude that this difference may be closely related to the occurrence and development of OA disease Line 391-395. Since each person has a unique microbiome, this statement may be too strong. And patient's race may affect and they never mentioned the geography of their samples or underlying health conditions other than OA.

Line 40; For the experiment, authors use 48 fecal samples (OA, n=20; HC, n=28), and this data are insufficient to develop a biomarker as mentioned in the conclusion of their abstract. Authors need to tone it down to observation for future experiments.
Line 84-92- What is the age group of selected patients? and show the parallel age matching with control and test group

Line 103- What do you mean by third trimester? A pregnancy stage?. Did you only compare pregnant women?

Line 201-215 and fig 1. There is a shift of Anaerobes in OA group and highlight them with major genera and have a section in discussion to address why.

Line 250-287 and Fig.2 Cryptococcus, Histoplasma and Mucor are generally not a part of human microbiome as they are only in pathogenic stages in mostly immunocompromised humans. Please justify your claims here with citation to support them. The citation you show do not claim they are part of normal flora. Please explain possible scenarios that led to show their DNA in feces.

Line 345- What made this correlation investigation to address co-existence of three groups of microbes? Explain the reason and no previous virome and bacteriome relations were established in the normal flora.

Line 367- Classification of osteoarthritis based on multi-kingdom signatures; this is a too strong claim based on smaller samples size. If this is to be claimed, you need to find another independent group of OA patients and show them that they have the same abundance.

Line 391-395- Samples size and variability is not enough to predict this. You can mention that its an observation in the study.

Line 444-446: "Candida albicans was enriched in OA patients and it may serve as a novel biomarker to study the pathogenesis of OA caused by fungal infections". C. albicans is also a part of normal flora and did authors check for any current or previous microbial infections associated with patients? In addition, a considerable fraction of healthy individuals are naturally colonized by candida and this is not a sound discussion part.

Minor comments. Authors used the word "Enriched" in the OA subjects repeatedly and its not an appropriate word to show abundance in OA subject. Consider changing the word.

Staff Comments:

Preparing Revision Guidelines

Please return the manuscript within 60 days; if you cannot complete the modification within this time period, please contact me. If you do not wish to modify the manuscript and prefer to submit it to another journal, please notify me of your decision immediately so that the manuscript may be formally withdrawn from consideration by Microbiology Spectrum.

Chen et al, pursued a very interesting study of comparing gut microbiome of health subset and individuals having osteoarthritis using whole genome sequencing. They found no significant difference in terms of individual microbial population. However, they report significant difference in abundance of identified genera suggesting it would help for developing future biomarkers to identify people with osteoarthritis. This is a very novel idea even though it does not address how to achieve such prediction. They only looked at people already having osteoarthritis and purpose of prediction for future risk is not possible. In addition, their sample size is too small for most of the statistics to accurately predict and the claims based on these should be toned down as significant observations. Authors are not highlighting the patient population with age group. What they observed is a change in the abundance, not the species diversity/variation. The authors conclude that this difference may be closely related to the occurrence and development of OA disease Line 391-395. Since each person has a unique microbiome, this statement may be too strong. And patient's race may affect and they never mentioned the geography of their samples or underlying health conditions other than OA.

Line 40; For the experiment, authors use 48 fecal samples (OA, n=20; HC, n=28), and this data are insufficient to develop a biomarker as mentioned in the conclusion of their abstract. Authors need to tone it down to observation for future experiments.

Line 84-92- What is the age group of selected patients? and show the parallel age matching with control and test group

Line 103- What do you mean by third trimester? A pregnancy stage?. Did you only compare pregnant women?

Line 201-215 and fig 1. There is a shift of Anaerobes in OA group and highlight them with major genera and have a section in discussion to address why.

Line 250-287 and Fig.2 Cryptococcus, Histoplasma and Mucor are generally not a part of human microbiome as they are only in pathogenic stages in mostly immunocompromised humans. Please justify your claims here with citation to support them. The citation you show do not claim they are part of normal flora. Please explain possible scenarios that led to show their DNA in feces.

Line 345- What made this correlation investigation to address co-existence of three groups of microbes? Explain the reason and no previous virome and bacteriome relations were established in the normal flora.

Line 367- Classification of osteoarthritis based on multi-kingdom signatures; this is a too strong claim based on smaller samples size. If this is to be claimed, you need to find another independent group of OA patients and show them that they have the same abundance.

Line 391-395- Samples size and variability is not enough to predict this. You can mention that it's an observation in the study.

Line 444-446: “Candida albicans was enriched in OA patients and it may serve as a novel biomarker to study the pathogenesis of OA caused by fungal infections”. *C. albicans* is also a part of normal flora and did authors check for any current or previous microbial infections associated with patients? In addition, a considerable fraction of healthy individuals are naturally colonized by candida and this is not a sound discussion part.

Minor comments. Authors used the word “Enriched” in the OA subjects repeatedly and its not an appropriate word to show abundance in OA subject. Consider changing the word.

Reviewer #1 (Comments for the Author):

1. Improper use of italics or non-italic in the strain name.

Response: Thanks for pointing this out. We have corrected all taxonomic names and strain names in the revised manuscript, following your suggestion.

2. In this study, criteria for judging significance level should be unified.

Response: Thanks for this variable comment. In the revised manuscript, we have updated all results using a unified criterion for judging significance level. This criterion was added in the “Methods - Statistical analyses” section as follows: “*A p-value (for a single test) or q-value (for multiple testing) less than 0.05 was considered statistically significant.*”

3. The lack of the icon for OA and HC in Figure 1f and Figure 2e.

Response: Thanks for pointing this out. We have added the icons in these figures as suggested.

4. Figure 2d should be reconfirmed to describe relative abundance at the genus or family level.

Response: Thanks for pointing this out. Figure 2d was performed at the genus level. We have corrected this expression in the revised manuscript.

5. The criterion for clustering viral contigs into vOTUs should be the same in the context.

Response: Thanks for this professional comment. The viral contigs were clustered at 95% sequence similarity and over 70 coverage to generate the vOTUs. We have unified this method in the revised manuscript.

6. Figure 3i and Figure 3l overlapped.

Response: Thanks for pointing this out. We have separated these two subfigures in the revised manuscript as suggested.

7. The color of the same family should be the same in Figure 3c, 3j and 3k.

Response: Thanks for pointing this out. We have updated Figure 3j and 3k based on the bar plot, and this issue no longer exists in the revised manuscript.

8. The lack of the figure note for Figure 3l.

Response: Thanks for pointing this out. We have added the notes for Figure 3l in the revised manuscript as suggested.

9. How was sample size calculated?

Response: Thanks for pointing this out. In the revised manuscript, we have increased the sample size to 44 OA patients and 46 age- and body mass index-matched healthy controls, according to suggestions by all three reviewers. The new fecal samples (24 OA patients and 18 controls) were collected in the same hospital as the previous samples during the same period (Aug. 2019 – Dec. 2020) and stored in a -80°C freezer until DNA extraction and sequencing. We have updated all results and conclusions in the revised manuscript based on the new dataset.

Reviewer #2 (Comments for the Author):

The manuscript is well written and shows extensive analysis of the microbiome found in patients with Osteoarthritis (OA) fecal samples. However, the sampling and analysis methods made are not correct for describing microbial patterns for AO. Then the results and conclusions are not valid.

The objective is described as elucidating associations of the gut microbiome and virome with OA. However, the authors also mentioned biomarkers as conclusions. Therefore, they should clarify the intentions of this work since the establishment of biomarkers requires different confirmations.

Response: We appreciated the reviewer's insightful and constructive comments. In the revised manuscript, we have modified our results and clarified our conclusions accordingly to your professional suggestions, which we believe have improved the manuscript greatly. In particular, the sample size of the current study was increased to 44 OA patients and 46

age- and body mass index-matched healthy controls to reinforce our results. Please see below for the responses to your comments.

The authors show some flaws in technical criteria, including:

- They should avoid using "gut microbiome" since the whole sampling and analysis was based on fecal samples. They should use "fecal microbiome" instead. One can speculate that the fecal microbiome might also be present in the gut, but not that the whole gut microbiome is represented on the fecal sample.

Response: Thank you for your professional comment. We agree with the reviewer that the microbiome in fecal samples is not fully representative of the gut microbiome. In practice, however, most studies (e.g., ref #1-5 listed below) still use the metagenomic sequencing of fecal samples to characterize the gut microbiome due to difficulties in sampling the rest of the human gut. In this study, we also used "gut microbiome" instead of "fecal microbiome" to simplify the overall understanding, since we did not study the situation of other "non-fecal" gut microbes. In addition, following your suggestion, we have emphasized multiple times in the revised manuscript that the object of our study was fecal microbiota, in order to avoid misleading.

[1] Qin J, Li Y, Cai Z, et al. A metagenome-wide association study of gut microbiota in type 2 diabetes. *Nature*, 2012, 490(7418): 55-60.

[2] Jie Z, Xia H, Zhong S L, et al. The gut microbiome in atherosclerotic cardiovascular disease. *Nature communications*, 2017, 8(1): 1-12.

[3] Yan Q, Gu Y, Li X, et al. Alterations of the gut microbiome in hypertension. *Frontiers in cellular and infection microbiology*, 2017, 7: 381.

[4] Kuang Y S, Lu J H, Li S H, et al. Connections between the human gut microbiome and gestational diabetes mellitus. *GigaScience*, 2017, 6(8): gix058.

[5] Wang X, Yang S, Li S, et al. Aberrant gut microbiota alters host metabolome and impacts renal failure in humans and rodents. *Gut*, 2020, 69(12): 2131-2142.

- The authors should use "microbiome" to refer to bacteriome and mycobiome. However,

using "microbiome" in addition to mycobiome is confusing and incorrect.

Response: Thanks for this valuable comment. We have carefully checked the revised manuscript and avoided the wrong use of these words.

- Gut bacteriome analysis: It should be more accurate to use a complete prokaryotes database, for example, Silva or Greengenes. The authors are limiting the results to the known gut bacteriomes.

Response: Thanks for pointing this out. In this study, the gut bacteriome composition of fecal samples was profiled based on the Unified Human Gastrointestinal Genome (UHGG) database [1]. This database was, to our knowledge, the currently most comprehensive, high-quality reference genome collection which comprised over 200,000 nonredundant genomes from 4,644 gut prokaryotes. And more than 70% of the UHGG species were uncultured.

Besides, we didn't use the Silva or Greengenes as these databases were primarily used for the analysis of amplicon sequencing data.

[1] Almeida A, Nayfach S, Boland M, et al. A unified catalog of 204,938 reference genomes from the human gut microbiome. *Nature biotechnology*, 2021, 39(1): 105-114.

- Mycobiome: again, the fungal assignment is limited to a selection of 106 genomes. This made the study completely biased.

Response: Thanks for pointing this out. In this study, to construct an available human gut fungal genome catalogue, we searched the NCBI genome database and BioSample metadata and identified totaling 233 fungal genomes (representing 106 species) that were isolated or sourced from human feces and digestive tract samples. For each species, the other non-human-gut strains/genomes were also included, generating a final catalogue of 1,503 fungal genomes for follow-up analyses. Therefore, due to the lack of reference genomes, this catalogue was currently a relatively complete collection of the human gut fungi. In the revised manuscript, we have clarified these methods in section "Methods - Analyses of gut mycobiome" to avoid misleading.

In particular, the non-human-gut-derived species (e.g., environmental fungi or mushrooms) were not included to make our results more accurate.

- It is not explained how the metagenomes were assembled.

Response: Thanks for pointing this out. We have added the assembly method in the revised manuscript in section “Methods - Identification of viral sequences and analyses of gut virome” as follows: “*High-quality metagenomic reads were used for de novo assembly via MEGAHIT (26) with a broad range of k-mer sizes (--k-list 21,41,61,81,101,121,141).*”

- The cohort seems too short for this analysis and to support the conclusions. The variability in relative diversity profiles is high in all the analyses. Establishing a dysbiosis or microbiome profile from 20 patients is not possible. Then, a larger group of patients will be necessary to confirm the profile patterns. Or at least statistically proof that this sample is enough for some of the dysbiosis events.

Response: Thanks for pointing this out. In the revised manuscript, we have increased the sample size to 44 OA patients and 46 age- and body mass index-match healthy controls, according to suggestions by all three reviewers. The new fecal samples (24 OA patients and 18 controls) were collected in the same hospital as the previous samples during the same period (Aug. 2019 – Dec. 2020) and stored in a -80°C freezer until DNA extraction and sequencing. We have updated all results and conclusions in the revised manuscript based on the new dataset.

- Also, the authors made a Spearman correlation of the microbiome with the OA condition, but they did not clarify other potential correlations of these results. For example, might it be an influence of ethnicity? Other risk factors? Diet? Age? How can they conclude an association without knowing other potential drivers of the differential microbiome between the cohorts? The baseline cohort characteristics need to be established and discard significant differences among them to present valid results.

Response: Thank you for your constructive comment. In the revised manuscript, we added a new supplementary table (**Table S1**) to show the phenotypic and clinical characteristics

of OA patients and healthy individuals. Patients and controls were matched in their age (51.8 ± 12.8 vs. 50.9 ± 11.9 years, $p=0.799$) and body mass index (23.4 ± 3.0 vs. 23.7 ± 2.9 , $p=0.618$). However, the patients had a lower proportion of female subjects than the control group (22.7% vs. 39.1%, $p=0.114$) but not significant. The smoke and dietary habits were also not significantly different between the two groups. Regarding ethnicity, all subjects were Han Chinese from the same region of Guizhou province, China. In addition, according to your suggestion, for Spearman correlation analysis in the revised manuscript, we calculated the correlation coefficients between different microbes based on the relative abundances after adjusting for individuals' gender, age, and body mass index (see sections "Methods - Correlation network analysis" and "Results - Correlations among gut bacteriome, mycobiome, and virome" for detail).

Reviewer #3 (Comments for the Author):

Chen et al, pursued a very interesting study of comparing gut microbiome of health subset and individuals having osteoarthritis using whole genome sequencing. They found no significant difference in terms of individual microbial population. However, they report significant difference in abundance of identified genera suggesting it would help for developing future biomarkers to identify people with osteoarthritis. This is a very novel idea even though it does not address how to achieve such prediction. They only looked at people already having osteoarthritis and purpose of prediction for future risk is not possible. In addition, their sample size is too small for most of the statistics to accurately predict and the claims based on these should be toned down as significant observations. Authors are not highlighting the patient population with age group. What they observed is a change in the abundance, not the species diversity/variation. The authors conclude that this difference may be closely related to the occurrence and development of OA disease Line 391-395. Since each person has a unique microbiome, this statement may be too strong. And patient's race may affect and they never mentioned the geography of their samples or underlying health conditions other than OA.

Response: We appreciate the reviewer's constructive and insightful comments. In the revised manuscript, we have addressed your concerns by adding new samples, updating the results and figures/tables, and modifying the texts and discussions. Below is a summary of your comments:

For prediction analysis. We agree with the reviewer that the classification of disease and healthy subjects by their microbiota is not a prediction. We have revised and toned down the corresponding statements in the revised manuscript (see sections "Results - Classification of osteoarthritis based on multi-kingdom signatures" and "Discussion") according to your suggestion.

For sample size. We have increased the sample size to 44 OA patients and 46 age- and body mass index-matched healthy controls, according to suggestions by all three reviewers. The new fecal samples (24 OA patients and 18 controls) were collected in the same hospital as the previous samples during the same period (Aug. 2019 – Dec. 2020) and stored in a -80°C freezer until DNA extraction and sequencing. We have updated all results and conclusions in the revised manuscript based on the new dataset.

For demographic characteristics. We have added a new supplementary table (Table S1) in the revised manuscript to show the phenotypic and clinical characteristics of OA patients and healthy individuals. Patients and controls were matched in their age (51.8 ± 12.8 vs. 50.9 ± 11.9 years, $p=0.799$) and body mass index (23.4 ± 3.0 vs. 23.7 ± 2.9 , $p=0.618$). However, the patients had a lower proportion of female subjects than the control group (22.7% vs. 39.1%, $p=0.114$) but not significant. The smoke and dietary habits were also not significantly different between the two groups. Regarding ethnicity and geography, all subjects were Han Chinese from the same region of Guizhou province, China.

For line 391-395. In the revised manuscript, these sentences were changed into "*Our results showed that, though the species diversity had not statistically differences, there were significant differences in the abundance of 279 bacterial species, 10 fungal species, and 627 viruses between two groups. Identification of these gut microbial signatures may facilitate further mechanistic studies of OA and related diseases.*" to clarify our results and avoid overstatement.

Line 40; For the experiment, authors use 48 fecal samples (OA, n=20; HC, n=28), and this data are insufficient to develop a biomarker as mentioned in the conclusion of their abstract. Authors need to tone it down to observation for future experiments.

Response: Thanks for this valuable comment. In the revised manuscript, we have increased the sample size to 44 OA patients and 46 healthy controls and updated all results and conclusions in the revised manuscript based on the new dataset. Also, we have toned down the descriptions of our results and conclusions in the revised manuscript based on your suggestion.

Line 84-92- What is the age group of selected patients? and show the parallel age matching with control and test group

Response: Thanks for pointing this out. In this study, patients and controls were matched in their age (51.8 ± 12.8 vs. 50.9 ± 11.9 years, $p=0.799$). Following your suggestion, this information was added in Table S1 and highlighted in sections “Methods - Ethics statement, subjects, and sample collection” and “Results - Biodiversity, phylogenetic and functional compositions of the gut bacteriome” in the revised manuscript.

Line 103- What do you mean by third trimester? A pregnancy stage?. Did you only compare pregnant women?

Response: Sorry for this error. We have deleted this sentence in the revised manuscript.

Line 201-215 and fig 1. There is a shift of Anaerobes in OA group and highlight them with major genera and have a section in discussion to address why.

Response: Thanks for pointing this out. In the revised manuscript, we have removed the discussion of the shift of Anaerobes in OA patients, as the results of *Proteobacteria* (7.8% vs. 3.3% in patients vs. controls, $q=0.018$) and *Escherichia* ($q>0.05$) were no longer particularly prominent based on the new cohort.

Line 250-287 and Fig.2 Cryptococcus, Histoplasma and Mucor are generally not a part of human microbiome as they are only in pathogenic stages in mostly immunocompromised

humans. Please justify your claims here with citation to support them. The citation you show do not claim they are part of normal flora. Please explain possible scenarios that led to show their DNA in feces.

Response: Thanks for this valuable comment. Following your suggestion, we have added a discussion in the revised manuscript (3rd paragraph in section “Discussion”) as follows: “Fungi such as *Cryptococcus* and *Mucor* are not a resident of the gut microbiota, but may link to pathogenic stages in mostly immunocompromised humans (58, 59).”

Line 345- What made this correlation investigation to address co-existence of three groups of microbes? Explain the reason and no previous virome and bacteriome relations were established in the normal flora.

Response: Thanks for pointing this out. In this section, we investigated the associations among three groups of OA-associated signatures based on a Spearman correlation coefficient analysis. We identified a total of 11,082 co-abundance relationships between 200 bacterial species and 418 viruses at the threshold of absolute correlation coefficient >0.6. However, no strong correlation was found between gut fungi and bacteria/viruses. The relationships between gut virome and bacteriome were previously disease-based studied such as colorectal cancer and inflammatory bowel disease [1-2], but it is still a lack of adequate study in normal individuals. The gut viruses and bacteria can be linked by 1) host-virus relationships, 2) some viruses and bacteria have potential symbiotic or resistant relationships, and 3) other unknown reasons. In particular, the centrality of some viruses in correlation networks, including members of *Myoviridae*, *Siphoviridae*, and *crAss-like* in the current study (listed in Figure 4c in the manuscript), may imply a potential important role for these viruses in disease, which needs further study. On the other hand, some viruses seem to act on disease independently of bacteria, such as the members of *Microviridae* (this result was added in the revised manuscript in Figure 4d-e), and their roles in disease also need further study.

[1] Nakatsu G, Zhou H, Wu W K K, et al. Alterations in enteric virome are associated with colorectal cancer and survival outcomes. *Gastroenterology*, 2018, 155(2): 529-541.

[2] Clooney A G, Sutton T D S, Shkoporov A N, et al. Whole-virome analysis sheds light on viral dark matter in inflammatory bowel disease. *Cell host & microbe*, 2019, 26(6): 764-778.

Line 367- Classification of osteoarthritis based on multi-kingdom signatures; this is a too strong claim based on smaller samples size. If this is to be claimed, you need to find another independent group of OA patients and show them that they have the same abundance.

Response: Thanks for pointing this out. In the revised manuscript, we have increased the sample size to 44 OA patients and 46 healthy controls and updated all results and conclusions in the revised manuscript based on the new dataset. Besides, to address the reviewer's concern, we trained a random forest model using the relative abundances of the OA-associated gut microbial signatures in the previous samples (20 OA vs. 28 HC) and tested the model in the newly sequenced samples (24 OA vs. 18 HC). The result was shown in the following supporting figure. We found that, in new samples, the model still

obtained a high performance (AUC = 0.905) in classifying the OA patients and healthy controls. This result supported the accuracy of OA classification.

Line 391-395- Samples size and variability is not enough to predict this. You can mention that its an observation in the study.

Response: Thanks for this valuable comment. In the revised manuscript, we have increased the sample size to 44 OA patients and 46 healthy controls and updated all

results and conclusions in the revised manuscript based on the new dataset. These sentences in previous Line 391-395 have been changed into “*Our results showed that, though the species diversity had no statistical difference, there were significant differences in the abundance of 279 bacterial species, 10 fungal species, and 627 viruses between two groups. Identification of these gut microbial signatures may facilitate further mechanistic studies of OA and related diseases.*” to clarify our results and avoid overstatement.

Line 444-446: "Candida albicans was enriched in OA patients and it may serve as a novel biomarker to study the pathogenesis of OA caused by fungal infections". C. albicans is also a part of normal flora and did authors check for any current or previous microbial infections associated with patients? In addition, a considerable fraction of healthy individuals are naturally colonized by candida and this is not a sound discussion part.

Response: Thanks for this insightful comment. We agree with the reviewer that *Candida albicans* is a part of the normal flora in the human gut. Based on the newly-sequenced samples, we found that the *Candida* genus was more abundant in the OA patients compared with healthy controls (11.6% vs. 5.9%, $p=0.008$), along with several *Candida* species such as *Candida parapsilosis* GCA_000182765 and *Candida* sp. LD148194 GCA_001005365. However, the *Candida albicans* was no longer significantly different in abundance between the two groups. Thus, we have removed the inaccurate discussions about *C. albicans* in the revised manuscript to clarify our findings.

About infection, both OA patients and healthy subjects were those who had not been infected recently and were not taking antibiotics and antifungals within 4 weeks before sampling.

Minor comments. Authors used the word "Enriched" in the OA subjects repeatedly and it's not an appropriate word to show abundance in OA subject. Consider changing the word.

Response: Thanks for pointing this out. To avoid excessive repetition, we have changed this expression into “more abundant in OA subjects”, “higher in OA subjects”, or others in the revised manuscript, unless it's necessary.

September 13, 2022

Prof. Wukai Ma
The Second Affiliated Hospital of Guizhou University of Traditional Chinese Medicine
Department of Rheumatology and Immunology
Feishan Street No.83
Guiyang, Guizhou
China

Re: Spectrum01711-22R1 (Characterizations of the gut bacteriome, mycobiome, and virome in patients with osteoarthritis)

Dear Prof. Wukai Ma:

Your manuscript has been accepted, and I am forwarding it to the ASM Journals Department for publication. You will be notified when your proofs are ready to be viewed.

Sincerely,

S. Wesley Long
Editor, Microbiology Spectrum

Journals Department
Supplemental Dataset4: Accept
Supplemental Dataset9: Accept
Supplemental Dataset2: Accept
Supplemental Dataset8: Accept
Supplemental Dataset1: Accept
Supplemental Dataset5: Accept
Supplemental Dataset3: Accept
Supplemental Dataset7: Accept
Supplemental Dataset6: Accept